# Microbial Enrichment Techniques on Syngas and CO_2_ Targeting Production of Higher Acids and Alcohols

**DOI:** 10.3390/molecules28062562

**Published:** 2023-03-11

**Authors:** Styliani Konstantinidi, Ioannis V. Skiadas, Hariklia N. Gavala

**Affiliations:** Department of Chemical and Biochemical Engineering, The Technical University of Denmark, Soeltofts Plads 228A, 2800 Kgs. Lyngby, Denmark

**Keywords:** syngas, carbon dioxide, butyric acid, caproic acid, butanol, microbial enrichment, metabolic potential, microbial communities, microbial consortia, co-culture

## Abstract

(1) Background: Microbial conversion of gaseous molecules, such as CO_2_, CO and H_2,_ to valuable compounds, has come to the forefront since the beginning of the 21st century due to increasing environmental concerns and the necessity to develop alternative technologies that contribute to a fast transition to a more sustainable era. Research efforts so far have focused on C1–C2 molecules, i.e., ethanol and methane, while interest in molecules with higher carbon atoms has also started to emerge. Research efforts have already started to pay off, and industrial installments on ethanol production from steel-mill off-gases as well as methane production from the CO_2_ generated in biogas plants are a reality. (2) Methodology: The present study addresses C4–C6 acids and butanol as target molecules and responds to how the inherent metabolic potential of mixed microbial consortia could be revealed and exploited based on the application of different enrichment methods (3) Results and Conclusions: In most of the enrichment series, the yield of C4–C6 acids was enhanced with supplementation of acetic acid and ethanol together with the gas substrates, resulting in a maximum of 43 and 68% (e-mol basis) for butyric and caproic acid, respectively. Butanol formation was also enhanced, to a lesser degree though and up to 9% (e-mol basis). Furthermore, the microbial community exhibited significant shifts depending on the enrichment conditions applied, implying that a more profound microbial analysis on the species level taxonomy combined with the development of minimal co-cultures could set the basis for discovering new microbial co-cultures and/or co-culturing schemes.

## 1. Introduction

Continuously expanding demands for energy supply, concerns about the environmental consequences of increasing CO_2_ emissions, as well as depleted fossil reserves, have brought alternative feedstock and means to generate energy, chemicals and material to the forefront during the last decades [1,2,3]. Besides organic residual material, i.e. lignocellulosic biomass, wastes and wastewater, much attention is also given to the possibility of valorizing “challenging” (not easily biodegradable) biomass, for example, sludge from wastewater treatment plants, used and non-recyclable plastic and elastic material, through gasification and subsequent chemical and/or biological conversion of the generated synthesis gas (syngas) [4,5,6]. Supplementation of industrial off-gases, which are rich in CO and CO_2_, with reducing power in the form of renewable electricity and/or H_2_, is another possibility that is currently very much investigated worldwide [7]. 

During the last decades, there has been explosive growth in research activities on syngas and CO_2_ biological conversion to energy carriers such as methane and ethanol, or chemical precursors such as acids and alcohols with higher carbon atoms [7]. Most of the activities focus on applying axenic or co-cultures, while an increasing interest in the potential of Mixed (undefined) Microbial Consortia, MMC (also referred to as microbial communities) as biocatalysts is also observed. Microbial communities present a series of benefits derived from their inherent microbial diversity and functional redundancy; however, their high complexity also constitutes one of their major limitations as they often result in decreased product selectivity, requiring further development of community management strategies. So far, application of MMC has been exceptionally promising for the production of methane and acetic acid [8,9,10,11,12], while it has also shown high potential for ethanol production, especially when the medium is supplemented with acetic acid [11,12,13]. Production of molecules with higher carbon atoms is mainly investigated in processes based on pure, wild or engineered microbial strains or co-cultures [14,15,16,17], while there is an emerging interest also in MMC [18,19,20,21]. Overall, research on producing C4-C6 acids and corresponding alcohols from gaseous substrates is still at an early stage, as depicted in Figure 1, which shows the number of Web of Science publications since 2006 with respect to different products from syngas. While ethanol production from syngas has already reached the industrial stage, there is considerably less research effort devoted to other products, with an increasing trend though.

Generation of acetic acid and ethanol from syngas and CO_2_/H_2_ is a common metabolic option. On the contrary, only a few microbial species have the possibility to directly generate medium-chain fatty acids and alcohols from C1 substrates via an extended Wood-Ljungdahl pathway, where 2 acetyl-CoA molecules are elongated to butyryl-CoA while assimilating electrons. Butyryl-CoA can then be converted to butyric acid, accompanied by the generation of ATP, or reduced to butanol using electrons [22]. Alternatively, the butyric acid generated may be reduced to butanol [22] in a metabolic scheme resembling acetone-butanol-ethanol (ABE) fermentation [23]. There is evidence that butyric acid formation from C1 compounds by a single species takes place at acidic pH [22], and it is mostly a protection mechanism of the cells to alleviate inhibition caused by acidic environments. Butanol formation is less energetically favorable as no energy conservation in the form of ATP formation via substrate-level phosphorylation is associated with butanol production. 

Butyric acid may also be generated in a two-step process through cross-feeding microbial interactions. During this process, a microbial species able to perform reversed β-oxidation uptakes acetic acid and ethanol to form butyric acid with concomitant generation of H_2_, in a metabolic scheme commonly known as chain elongation. In such a two-step process, acetogenesis and chain elongation reactions are compartmentalized in different microbial species or carried out by different bacteria [15]. Caproic acid can also be produced via chain elongation, using butyric acid and ethanol as the starting carbon and electron donors. Therefore, co-culturing two microbial species on C1 gas substrates under a cross-feeding scheme, where the second species utilizes the metabolic products of the first, is another promising alternative to single-cell approach [24,25]. This type of mutualistic interaction is naturally anticipated to be the dominant mechanism of C4 and C6 organic acids formation in MMC environments at pH values that are close to neutral, and this is the focus of the current study. 

The complex nature and high microbial diversity of MMC, which challenges their specificity towards one product, is an advantage when it comes to the discovery of novel strains or co-cultures targeting a specific biological conversion. Besides efforts in applying MMC to produce compounds with higher carbon atoms from syngas and CO_2_/H_2_, and in the frame of an alternative research direction, it is important to also take advantage of the diverse nature and enrich our state-of-the-art knowledge with new and potentially even more efficient microbial strains than those that are widely known and investigated nowadays. The research question in this case is what selective pressure should be applied to reveal the inherent potential of a MMC for a specific bioconversion. This is the first step to generating and characterizing novel and efficient microbial co-cultures regarding their microbial composition as well as metabolic potential. While molecular tools and methods have enabled us to advance in respect to characterizing MMC in-depth [26], differentiating the selective pressure could eventually lead to spotting promising microbial species or co-cultures that thrive under the conditions applied. 

In the present study, different selective pressures were investigated to enrich MMC in microbial strains that can produce elongated products such as butyric and caproic acids as well as butanol from syngas and CO_2_. This is a preliminary step to an effort focusing on unlocking and expressing the metabolic potential of a microbial consortium and enriching microbial consortia in strains that can produce the target compounds. Carrying out microbial enrichments via batch transfers on syngas as a sole carbon and energy source has been very efficient in enriching strains able to thrive on syngas and generate C1–C2 products, i.e., methane, acetic acid and ethanol [13,27]; they are not too efficient, though, in generating elongated products or higher alcohols. The main hypothesis in the frame of the current study was that supplementing the gas substrates with precursors of chain elongation, i.e., acetic acid and ethanol, would boost chain elongation, while supplementation of butyric acid with reducing power in the form of CO or H_2_ might trigger production of butanol and/or caproic acid. As supplementing butyric acid with syngas or CO_2_/H_2_ did not really activate the microbial consortium, another series of experiments was carried out with acetic acid and ethanol as substrates for growth in the first step. Reducing power in the form of CO or H_2_ was added after growth was resumed. The latter was based on the hypothesis that, as butanol production from butyric acid alone cannot support microbial growth, supplementation of reducing power at a later stage may trigger reduction at a higher degree than providing the gaseous electron donors in the beginning of the batches. The strategies mentioned above were tested with syngas and a mixture of CO_2_ and H_2_ to draw conclusions on the effect of CO compared to H_2_. The application potential of the enrichment techniques on both syngas and CO_2_ as substrates was also evaluated. The microbial enrichments were evaluated and compared based on the final products distribution. Finally, 16sRNA sequencing was conducted to reveal the degree of the enrichments’ differentiation.

## 2. Results

The experimental protocol followed in Section 2.1, Section 2.2 and Section 2.3 was based on microbial enrichment tests performed under different conditions in terms of acid/alcohol and gas addition. Each enrichment series was performed through successive batch tests conducted by transferring a part of the mixed liquor from one batch vial at the end of its cycle to another. A detailed description is given in Section 4.3.

### 2.1. 1st Experimental Series of Microbial Enrichments on Syngas and CO_2_/H_2_ Supplemented with Acetic Acid and Ethanol

The 1st experimental series is based on the addition of acetic acid and ethanol to promote chain elongation. Syngas and a gas mixture of CO_2_/H_2_ were added as gas substrates, while enrichment with a gas mixture of N_2_/CO_2_, where no reducing equivalents were added, served as the control. A detailed description is given in Section 4.3.

The products distribution in e-mol yield is presented in Figure 2 for all transfers, while Table 1 shows the final concentrations of major end-products obtained after the last transfer, as well as electron recovery. As shown in Table 1 and can also be derived from Figure 2, all enrichments reached a high electron recovery in measured products. The enrichments on syngas (Figure 2a) and N_2_/CO_2_ (Figure 2b) were successful regarding the formation of caproic acid (with e-mol yields of 37 and 41%, respectively) and butyric acids (an e-mol yield of 40% in both). Butanol was also produced at a significantly lower yield though compared to C4 and C6 acids (e-mol yields of 7.1 and 5.4, respectively). Interestingly, the enrichment on CO_2_/H_2_ (Figure 2c) exhibited lower yields in elongated products (22 and 3% for C4 and C6 acids, respectively) and a comparable yield of butanol (8.8%), with acetic acid being the main metabolite of these series (an e-mol yield of 61%). 

### 2.2. 2nd Experimental Series of Microbial Enrichments on Syngas and CO_2_/H_2_ Supplemented with Butyric Acid

The 2nd experimental series was based on performing enrichments with supplementation of butyric acid with reducing power in the form of CO or H_2_ to investigate whether adding butyric acid would trigger the production of butanol and/or caproic acid. The enrichments of this experimental series were performed at two pH values, 5.5 and 6.5, as described in Section 4.3.2. 

Table 2 shows the concentration and net production of major compounds and targeted end-products measured at the end of the first phase (transfer 0) of the microbial enrichments. During all experiments supplemented with butyric acid, the production of acetic acid and butyric acid was observed, indicating that a fraction of the gas substrates and/or produced acetic acid was converted to ethanol, which was subsequently fully consumed towards the formation of butyric acid. However, the amounts of butyric acid produced were small, while no production of caproic acid was observed. Butanol was detected in traces in some transfers. Interestingly, in the experiments with CO_2_/H_2_ at both pH values, net consumption of butyric acid was observed with no equivalent amounts of butanol or caproic acid produced. This, combined with the low electron recovery, may imply the generation of a product that was not detected. Based on the low recovery of C4 and C6 acids and butanol, the 2nd experimental series was discontinued. An explanation for the low efficiency could be that the amount of gas supplied could not support further elongation of butyric acid to form caproic acid, while reduction to butanol alone could not proceed as it does not result to energy conservation for the cell at the amount needed to support growth. These observations led to the 3rd series of experiments, where reducing power in the form of CO and H_2_ was provided after growth was resumed as described in Section 4.3.2. 

### 2.3. 3rd Experimental Series of Microbial Enrichments on Acetic Acid and Ethanol Supplemented with Reducing Power in the Form of CO or H_2_ after Growth Has Resumed

Supplementation of butyric acid with syngas or CO_2_/H_2_ did not really activate the microbial consortium as described in the previous section. Therefore, the 3rd experimental series was carried out with acetic acid and ethanol as substrates for growth in the first step, and addition of reducing power in the form of CO or H_2_ after growth was resumed. This was to investigate whether supplementation of reducing power at a later stage might trigger reduction at a higher degree than providing the gaseous electron donors in the beginning of the batches.

Products distribution in e-mol yield is presented in Figure 3 for all transfers, while Table 3 shows the final concentrations of major end-products obtained after the last transfer, as well as electrons recovery for the 3rd series of experiments.

The enrichments supplemented with H_2_ and CO (Figure 3a) were successful regarding the formation of caproic and butyric acids (with e-mol yields of 40 and 32%, respectively), while butanol was also produced at a much lower e-mol yield of 2%. On the other hand, the enrichments supplemented with H_2_ favored particularly the formation of caproic acid with an e-mol yield of 59%, the highest yield for C6 acid among all enrichments in this study. Butyric acid was the second major product at an e-mol yield of 29%. Butanol was also formed to a much lesser degree, with an e-mol yield of 1.3%. So, chain elongation was mostly favored when supplying H_2_ rather than a mixture of H_2_ and CO. 

### 2.4. Microbial Characterization of Enriched Cultures

Figure 4 illustrates the evolution of the microbial composition of the 1st and 3rd experimental series compared to the initial inoculum (Syn-T0) at genus level. 

The initial inoculum was characterized by an even distribution of different genera, with the most dominant being *Desulfovibrio*, *Proteiniphilum*, *Thermobrachium*, and *Clostridium sensu stricto 12*, accounting for approximately 25%, 20%, 20% and 15% of the total reads mapped, respectively. Other genera such as *Bacteroides*, *Clostridium sensu stricto 1*, *Fonticella*, *Methanobacterium*, *Paraclostidrium*, *Sporomusa* and *Sporosarcina* were also represented in smaller amounts. However, the composition of the enriched cultures shifted considerably after three sequential transfers under different enrichment conditions.

The sequential transfers led to the enrichment of predominantly *Clostridium*, *Desulfovibrio*, *Proteiniphilum* and *Sporanaerobacter* spp. in varying amounts depending on the enrichment series. *Clostridium* spp. may be associated with a range of metabolic activities, including fermentative, hydrogenotrophic, carboxydotrophic and chain-elongating activity [28,29]. Similarly, *Sporanaerobacter* spp., also belonging to the order Clostridiales, exhibits a fermentative metabolism and have been repeatedly observed in chain elongation processes [30,31]. On the other hand, *Proteiniphilum* spp. are strictly associated with proteolytic activity, typically growing on amino-acid- or peptide-rich carbon sources. This suggests that they were feeding on the yeast extract or on cell debris present in the broth [32]. *Desulfovibrio* spp. are characterized by their respiratory metabolism, using sulfate and other sulfur compounds as terminal electron acceptors. However, most *Desulfovibrio* spp. are also able to oxidize a variety of organic compounds such as ethanol or lactate into acetate [33]. This would be consistent with the considerable supplementation of ethanol as substrate during these enrichments. Other genera present in the enriched cultures and worth noting for their fermentative metabolism may include *Bacillus* spp. and *Dysgonomonas* spp. [34,35]. 

Both experimental series resulted in characteristic microbial community compositions. However, these presented different trends. While the 1st experimental series was enriched mainly with *Clostridium* spp., the latter were poorly represented in the 3rd experimental series. Instead, the 3rd experimental series presented a higher relative abundance of *Desulfovibrio*, *Proteiniphilum* and *Sporanaerobacter* spp. This indicates that, despite the microbial activity being dominated by chain elongation in all cases, the timing of the addition of the gaseous substrates had a significant impact on the microbial community structure.

## 3. Discussion

During the 1st series of experiments, the initial pH for all batches was set at a value of 6.5 to allow for chain elongation in the beginning. Throughout the duration of each batch, the pH was reduced in all enrichments due to the formation of acidic products. The final pH range was between 5.2 and 5.6. Low yields of butanol could be partly attributed to the relatively high pH values in the beginning of the batches, as alcohol formation is favored at pH values lower than 5.0 [36,37]. It is important to emphasize, though, that the addition of acetic acid and ethanol did trigger butanol production compared to the inoculum activation on syngas, where the major products detected were acetic and butyric acids accompanied by lower amounts of propionic and isovaleric acids, and no production of caproic acid or butanol at all (Appendix A).

Perez et al. [16] studied the reduction of short-chain carboxylic acids, including butyric acid, into their corresponding alcohols in axenic cultures of *Clostridium ljungdahlii* and *Clostridium ragsdalei*. They concluded that eliminating CO and supplying electrons only via H_2_ would be beneficial for the alcohol formation. The results of that study are not directly comparable with those of the present study since those experiments run with continuous gas supply and at a constant pH of 5.5. However, the same trend was not observed since butanol yield was comparable in the batches with syngas and CO_2_/H_2_ mixture. One major difference, though, is that in the present study, the microbial consortium has the potential to perform chain elongation, as proved by the high yields of caproic acid, while in the study of Perez et al. no chain elongation activity was present. Therefore, the increased yields of butyric and caproic acid in the experiments with syngas compared to these with CO_2_ and H_2_ in the first series of experiments are due to a higher yield of the ethanol as an intermediate in the chain elongation process.

Vasudevan et al. [19] studied the conversion of dilute ethanol and acetic acid to chain elongation products by mixed microbial consortia in a continuous reactor. Interestingly, they obtained a much higher butyric acid yield compared to caproic acid, while in the current study, the yields of C4 and C6 acids were comparable. One explanation for this difference could be attributed to the much higher concentrations reached in the study of Vasudevan et al. as this could have a selectively adverse effect on the sequential chain elongation of butyric acid to caproic acid. This is also in agreement with the hypothesis mentioned by Vasudevan et al. and is worthy of further investigation. On the other hand, enhanced yield of butyric acid compared to caproic acid was also obtained in the study of He et al. [38]. Interestingly, the concentrations achieved in that study were at the same order of magnitude as those obtained in the current study, and therefore increased products concentrations could not potentially be the explanation for this discrepancy. In this case, high pH values in the study of He et al. might have favored the production of acetic instead of ethanol, as also supported by the elevated concentration of acetic acid. Therefore, the chain elongation of butyric acid could not proceed further due to the lack of ethanol. It should be noted also that the mode of operation is critical for the degree of chain elongation obtained. For example, a high dilution rate for CSTR-based enrichments could prevent chain elongation due to either washing-out of microbial species with low specific growth rate or time-limitations for the chain elongation reactions to occur, favoring thus butyric over caproic acid formation.

The enrichments grown on syngas and CO_2_/N_2_ were more efficient in chain elongation compared to the series grown on CO_2_/H_2_, implying that the latter favors the generation of acetic acid. The low net e-mole yield of acetic acid in the experiment with N_2_/CO_2_ in the headspace can be explained by homoacetogenesis on CO_2_ added and H_2_ generated during chain elongation. On the other hand, the e-mole yield of acetic acid in the enrichments with CO_2_/H_2_ cannot be attributed solely to homoacetogenesis from CO_2_ and H_2_. Therefore, ethanol oxidation might have contributed to acetic acid formation. Ethanol oxidation has been reported in several studies [39,40,41], and the conditions that favor it during gas fermentation processes certainly deserve further investigation. 

During the 3rd series of experiments, supplying H_2_ favored mostly chain elongation over supplying a mixture of H_2_ and CO. One possible explanation for this is that H_2_ boosted acetic acid reduction to ethanol, which subsequently also boosted chain elongation to butyric and caproic acids. It is known that supplying CO favors the generation of reduced products, such as ethanol, in the Wood-Ljungdahl pathway by acetogens. This subsequently also boosts chain elongation. However, the results of the present study imply that the addition of H_2_ is preferable for promoting acetic acid reduction at a later stage. An additional explanation could be that, during chain elongation reactions, less H_2_ is generated per mole ethanol in the case of caproic acid compared to butyric acid. Therefore, the addition of H_2_ favors elongating butyric acid rather than acetic acid when the former is present. Moreover, the strategy applied in the 3rd experimental series resulted in a higher degree of utilization of the reducing equivalents added in the form of liquid (ethanol and acetic acid) and gaseous substrates (CO and/or H_2_) towards products with higher carbon atoms compared to the strategy applied in the 1st series. This is depicted clearly in Figure 5, where the distribution of net products (compounds with a positive e-mol yield) is shown. Furthermore, acetic acid is not among the net products of the 3rd series, contrary to the observations made during the 1st series, and especially the enrichments on CO_2_/H_2_ and to a lesser degree on N_2_/CO_2_. 

Conclusively, enriching mixed microbial consortia on syngas and CO_2_ in batch mode with supplementation of acetic acid and ethanol as precursor molecules resulted in enhanced chain elongation, favoring to a high extent caproic acid formation. Butanol formation was also observed, to a much lower extent though compared to C4 and C6 acids. On the other hand, supplying acetic acid and ethanol resulted in considerable butanol formation when considering that activation of inoculum on syngas did not yield any. The addition of reducing equivalents in the form of CO and/or H_2_ after growth was resumed did not trigger butanol formation; however, it did trigger caproic acid formation, especially in the case of H_2_ as an electron donor. Furthermore, it has been shown (Section 2.4) that the enrichment itself, as well as the differentiation of enrichment strategies, affected the microbial composition significantly. The latter implies that a more profound microbial analysis on the species-level taxonomy as well as metabolic activities combined with the development of minimal microbial co-cultures, could set the basis for discovering new microbial co-cultures and/or co-culturing schemes.

The present study also resulted in interesting observations that could trigger future research directions. Supplementation of the 1st experimental series with butyric acid could trigger elongation and reduction, while comparison of the enriched microbial cultures at species taxonomic level could reveal which species are responsible for the major activities observed. Indications of ethanol oxidation reactions require further elucidation on the conditions that may trigger alcohols oxidation to their respective acids, as reduced alcohols yields might as well be due to oxidation reactions. 

## 4. Materials and Methods

### 4.1. Inoculum Source, Pre-Treatment, and Activation

The inoculum used for the enrichment experiments consisted of a combination of two different types of anaerobic sludge. The first type was collected from Mølleåværket wastewater treatment plant in Kongens Lyngby, while the second one was obtained from a lab anaerobic digester fed with swine manure. These two types of sludge were mixed in equal volumes (50/50 *v*/*v*) and then subjected to heat pretreatment to suppress the methanogenic activity before cultivation. The heat pretreatment was carried out by placing the sludge mixture into a boiling water bath for 15 min while continuously flushing with N_2_.

Before starting the enrichments, the pretreated sludge was activated through incubation at 37 °C on gas substrates in five replicates. For this, 500 mL infusion flasks with an active volume of a 165 mL and an inoculum size of 18% *v*/*v* were used, which translates to 30 mL of inoculum in a 165 mL active volume. To each flask, 135 mL of growth medium were added, and the flasks were continuously flushed with N_2_ to ensure anaerobic conditions. After the inoculum was added, the pH was adjusted to 6.5 using 1 M HCl. The flasks were then sealed with rubber stoppers and aluminum crimps, and flushed with H_2,_ and CO and CO_2_ were added to a final pressure of approximately 1.8 bars. All gases used had purity above 99.9%. The composition of the gases in the headspace was 12.5 mmol of H_2_ (51%), 5.4 mmol of CO (22%) and 6.6 mmol of CO_2_ (27%). The fermentation flasks were incubated in the dark for approximately 10 days. After the activation period, as the five replicates performed similarly in terms of gas consumption and products formation, the contents of all five flasks were mixed. This mixture was used as the initial inoculum for the 1st and 2nd enrichment series. The inoculum for the 3rd series of enrichments was a mixture of liquor from the zero transfer of 1st and 3rd experiments of enrichment series 1 (syngas and CO_2_/H_2_ supplemented with acetic acid and ethanol).

### 4.2. Growth Medium

The growth medium used for the enrichment experiments was prepared by combining six stock solutions: macronutrients, yeast extract, sodium sulphate, vitamins, trace elements and phosphate buffer. The composition of the stock solutions is presented in Table 4. For the preparation of 1 L of media, the following volumes of stock solutions were added: 20 mL of macronutrients, 20 mL of yeast extract, 10 mL of sodium sulfate, 10 mL of vitamins and 10 mL of micronutrients. The phosphate buffer consisted of K_2_HPO_4_ and KH_2_PO_4_ salts, while the composition and volume added were calculated for each experiment separately based on equilibrium reactions and charge balance, and by considering the bases, acids and salts added to adjust the pH at the targeted value.

### 4.3. Microbial Enrichments

#### 4.3.1. Microbial Enrichment Methodology

Three experimental series of microbial enrichments were carried out under different conditions to boost the chain elongation for C4 and C6 acids, as well as butanol production. For the enrichment experiments, 300 mL serum vials with 100 mL active volume and an 18% inoculum size were used. The experiments were performed in triplicates, and the best-performing one (i.e., the vial with the higher production yield of the targeted products) was used as an inoculum for the following transfer. Each enrichment series was successively transferred 3 to 4 times, as depicted in Figure 6, as this is the number of transfers that has been proven adequate to reach a sufficiently enriched microbial consortium [13]. The growth medium used was fresh and kept in a glass bottle that was continuously flushed with N_2_ during the inoculation process. Fresh growth medium and the inoculum were added to the serum vials under N_2_ flashing. At this point, the respective intermediates (i.e., acetic acid and ethanol or butyric acid) of each enrichment strategy were also added to the serum vials, and pH was adjusted to the desirable value (see Table 5) with a 1 M HCl solution, and the vials were immediately sealed with rubber stoppers and aluminum crimps. Subsequently, gases were added, with the vials being flushed with the major gas, and the rest of the gases being added on top to a final pressure of 1.8 atm. Each vial was sampled once at the beginning and once at the end of each batch, while frequent gaseous samples were taken to monitor the progress of the fermentation. The measurements obtained from the initial and final sampling were used to calculate the electrons yields of products and substrates, as described in Section 4.3.3. Gaseous and liquid samples were analyzed for their headspace gas composition, pH, acids, and alcohols’ composition in the liquid phase. From the second transfer and onwards, OD was also measured to monitor the microbial growth.

#### 4.3.2. Experimental Strategy

Table 5 shows an overview of the microbial enrichment experiments. The experiments shared the same starting inoculum but were differentiated by the gas composition, initial pH, and addition of intermediates. Acetic acid and ethanol were added at concentration of 93 and 39 mM, respectively, for the 1st and 3rd series of experiments. A ratio of ethanol to acetic acid of 5:3 is considered appropriate for chain elongation to butyric acid based on the stoichiometry of the reaction (1); in the present study, a higher ratio was applied to also account for acetic acid production from the gases (syngas and CO_2_/H_2_) utilization. In the 2nd series of experiments, the starting concentration of butyric acid was set at 15 mM to allow for monitoring consumption/production while avoiding any inhibitory effect that might be the case with higher concentrations of butyric acid. The initial pH was adjusted to 6.5 for the 1st and 3rd series of experiments, while values of 6.5 and 5.5 were the initial point for the 2nd series of experiments supplemented with butyric acid. The 1st and 2nd series of experiments were run with either syngas or CO_2_/H_2_ addition as gaseous substrates at an initial pressure of 1.8 atm. A gas mixture of N_2_/CO_2_, where no reducing equivalents were added, served as the control for the experiment with CO and/or H_2_ in the headspace. The composition of the gas mixtures was calculated based on the syngas quality index, SQI, that expresses the e-mol per C-mol of the gas mixture (2) and was first introduced by Asimakopoulos et al. [42]. The SQI was set between 5 and 5.3 and 6 for chain elongation and butyric acid reduction to butanol, respectively, to match the e-mol per C-mol of the target products (i.e., 5, 5.33 and 6 for butyric acid, caproic acid and butanol, respectively). In all batches, CO initial partial pressure did not exceed 0.2 bar to avoid potential inhibition effects.
(1)$$5CH_{3}CH_{2}OH+3CH_{3}COOH\to 4CH_{3}CH_{2}CH_{2}COOH+2H_{2}+4H_{2}O$$
(2)$$SQI=\frac{2\cdot \left(\%CO+\%H_{2}\right)}{1\cdot \left(\%CO+\%CO_{2}\right)}$$
where 2 is the number of e-moles per mole of *CO* and *H*_2_ and 1 is the number of C-mol per mole *CO* and *CO*_2_.

#### 4.3.3. Calculations

At the end of the triplicate experiments (transfers) during the enrichments, the yield for each gaseous and liquid compound was calculated and expressed as the percentage of the e-mole of the compound produced or consumed per e-mole consumed according to the following equation: (3)$$Y_{i,j}\%=\frac{\Delta \left(e-mol\right)}{-\sum_{j}\Delta \left(e-mol\right)}\cdot 100$$
where *i* is the compounds with a net production (positive yields), *j* is the compounds that are consumed (negative yields) and Δ(e-mol) is the difference between the final and initial e-mol of the compound *i* or *j*. The e-mol is calculated as the number of electrons that 1 mole of the compound needs to transfer to become fully oxidized. For examples, *CO* and *H*_2_ have 2 e-mole per mole, acetic acid and ethanol have 8 and 12 e-mole per mole, respectively, while *CO*_2_ comes with zero e-mole per mole, as the carbon in *CO*_2_ is fully oxidized.

The recovery of electrons gives an estimation of the extent of identification and recovery of all products in the broth and was calculated based on Equation (4):(4)$$Recovery\;\%=\frac{\sum_{i,j}e-mole,end}{\sum_{i,j}e-mole,\;start}$$
where *e-mole*, *end* and *e-mole*, *start* is the number of e-mole calculated at the end and start of each batch, respectively.

The distribution of end products is calculated as follows:(5)$$Distribution\;\%=\frac{e-mole,\;end_{Mi}}{\sum_{i}e-mole,end}\cdot 100$$
where *Mi* is the major compounds with a net production (positive yield).

All enrichments series are characterized and compared in respect to the % electron yields of the target products, i.e., C4 and C6 acids and butanol, as well as the final concentration reached and net consumption of e-mole. 

### 4.4. Analytical Methods

Composition of the gas in the headspace was analyzed with a gas chromatograph (model 8610C, SRI Instruments, Bad Honnef, Germany), equipped with a thermal conductivity detector and two packed columns (a 6′ × 1/8″ Molsieve 13× column and a 6′ × 1/8″ silica gel column) connected in series. The gas chromatograph configuration was calibrated to quantify H_2_, CO, CO_2_, CH_4_ and O_2_ in the headspace. The temperature of the column was kept at 65 °C for 3 min followed by an increase of 10 °C/min until 95 °C was reached and a second increase of 24 °C/min up to 140 °C. The gas volume sampled from the experimental vials was 50 μL and was taken with a 0.25 mL gas-tight syringe (model 1750 SL, Hamilton, Boston, MA, USA). 

The concentrations of the organic acids and alcohols of the liquid samples were analyzed with a high-performance liquid chromatograph (Shimadzu, Torrance, CA, USA). The analytical method was calibrated for the determination of acetone, lactic acid, glycerol, formic acid, acetic acid, propionic acid, 1,3-PDO, iso-butyric acid, butyric acid, ethanol, iso-valeric acid, 1-butanol, caproic acid, hexanol, although only the major products are presented. The HPLC was equipped with a refractive index detector and an Aminex HPX-87H column (Bio-Rad, Copenhagen, Denmark) at 63 °C. The eluent was a solution of 12 mM H_2_SO_4_ at a flow rate of 0.6 mL/min. Liquid samples were centrifuged at 10,000 rpm for 10 min, the supernatant was acidified to a pH of 1–3 with 20% H_2_SO_4_, and filtered through syringe filters with a membrane filter of 0.22 µm pore size before injection.

The growth of the microbial biomass was monitored via OD measurements at 600 nm using a spectrophotometer (DR2800, Hach Lange, Ames, IA, USA), and the pH of the liquid samples was measured by a pH meter (PHM 210, meter lab, Radiometer Analytical, Villeurbanne, France). 

### 4.5. DNA Isolation, Amplicon Sequencing and Microbial Community Analysis

To analyze the microbial composition of the enrichments, samples were taken for DNA analysis. Samples were collected from the initial inoculum, which was the same for all conditions applied, as well as from the final enriched samples. Samples were taken from all vials of the triplicates. In total, 16 samples were taken, three for each of the 1st and 3rd series of experiments that yielded the most promising results, and one from the initial inoculum (generated as described in Section 4.1). 

The samples for DNA analysis were prepared by extracting 15 mL of each culture in falcon tubes and spinning them down (10,000 rpm for 10 min). DNA was isolated from all samples using the DNeasy Powersoil Kit (Qiagen, Vedbæk, Denmark), following the manufacturer’s recommendations. DNA samples were submitted to Macrogen Inc. (Seoul, Republic of Korea) for 16S rRNA amplicon library preparation and sequencing using the Illumina Miseq instrument (300 bp paired-end sequencing). The libraries were constructed according to the 16S Metagenomic Sequencing Library Preparation Protocol (Part #15044223, Rev. B) using Herculase II Fusion DNA Polymerase Nextera XT Index Kit V2. Regions V4-V5 of 16S rRNA gene were amplified with primers 515F (5′-GTGYCAGCMGCCGCGGTAA-3′) and 926R (5′-CCGYCAATTYMTTTRAGTTT-3′) [43].

Raw reads were primer-trimmed with cutadapt, discarding all untrimmed reads [44]. Low-quality tails were trimmed by a fixed length of 20 bases in forward reads and 60 bases in reverse reads. Trimmed reads were merged, quality-filtered and denoised using DADA2 within the Qiime2 pipeline [45]. Taxonomic assignment to ASVs was performed using classify-sklearn algorithm and a taxonomic classifier based on the MiDAS 4.81 database [45,46]. Downstream analyses were performed using the Phyloseq, ggpubr and R packages (Phyloseq version 1.28.0, Vegan version 2.5.6, ggpubr version 0.4.0, and R version 3.6.0) [47,48,49].

## Figures and Tables

**Figure 1 molecules-28-02562-f001:**
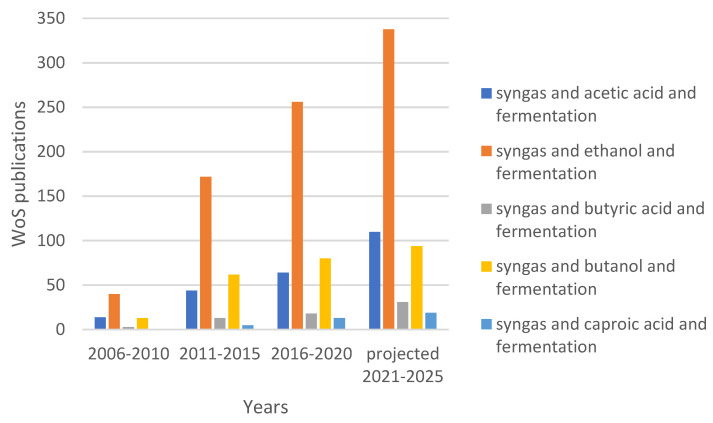
Number of publications related to C2–C6 products from syngas in the years 2006–2025 as found in Web of Science. The legend shows the keywords used for database search. The number of publications for the years 2021–2025 is calculated based on linear extrapolation of the number obtained for the years 2021–2022.

**Figure 2 molecules-28-02562-f002:**
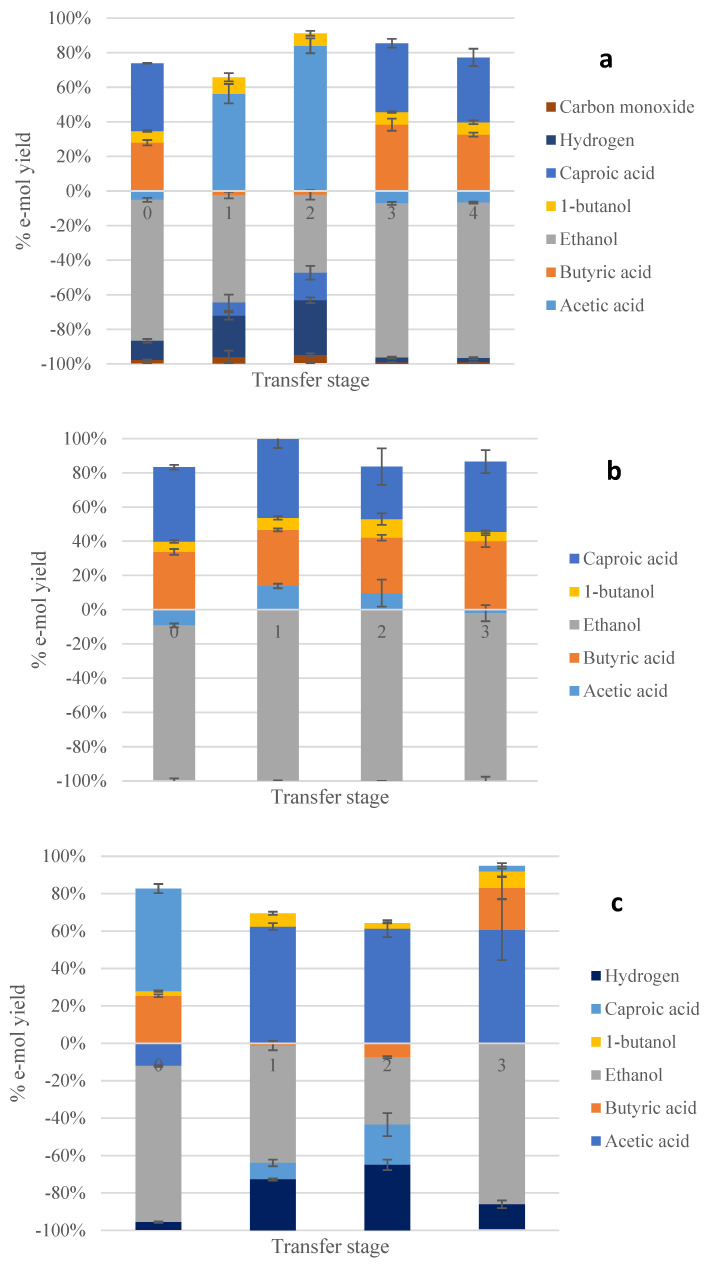
Electron yields for major compounds during the 1st experimental series with enrichments on syngas (**a**) CO_2_/N_2_, (**b**) and CO_2_/H_2_ (**c**) supplemented with acetic acid and ethanol. Positive stacked columns show the yield of products in e-mol basis for each batch transfer. Negative stacked columns indicate the source of e-mols for each batch transfer. Numbers “0–4” and “0–3” refer to the transfer stage during the enrichment process.

**Figure 3 molecules-28-02562-f003:**
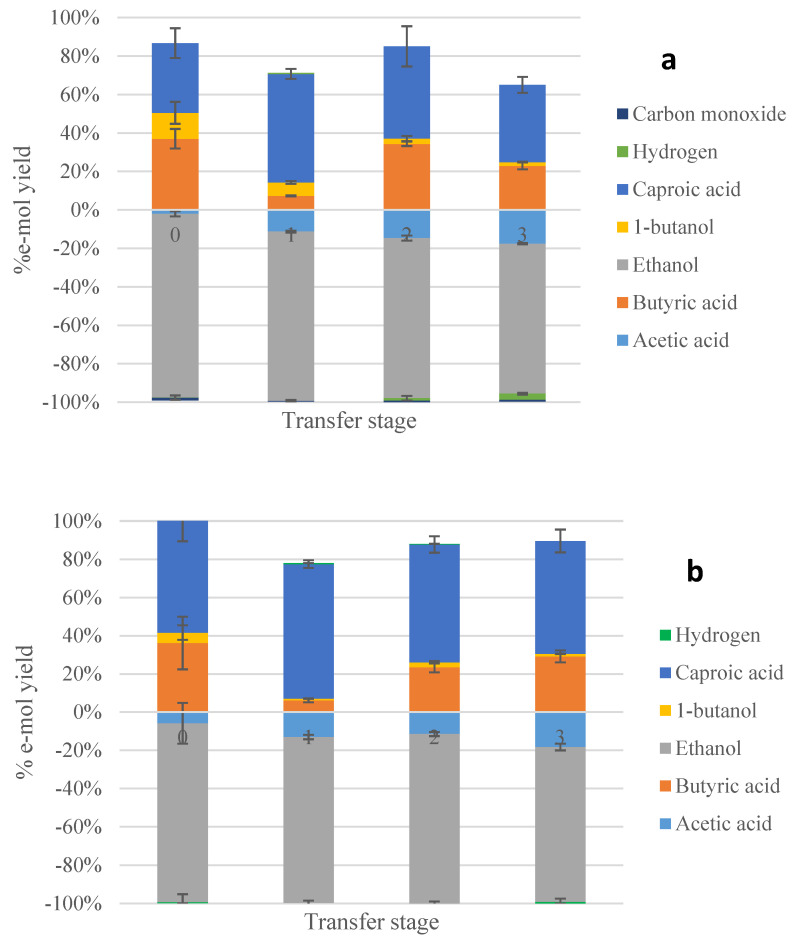
Electron yields for major compounds during the 3rd experimental series with enrichments on CO/H_2_ (**a**) and H_2_ (**b**) supplemented with acetic acid and ethanol. Numbers “0–3” refer to the transfer stage during the enrichment process.

**Figure 4 molecules-28-02562-f004:**
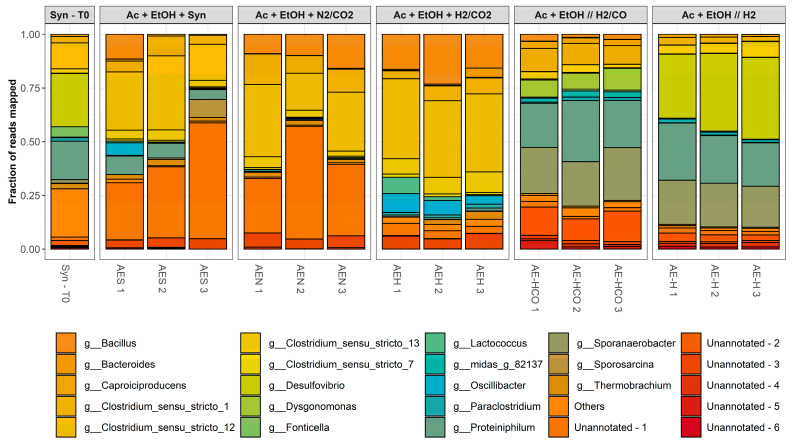
Relative abundance of reads mapped at genus level for the 1st and 3rd enrichment strategies. Samples were analyzed in triplicates from independent experiments.

**Figure 5 molecules-28-02562-f005:**
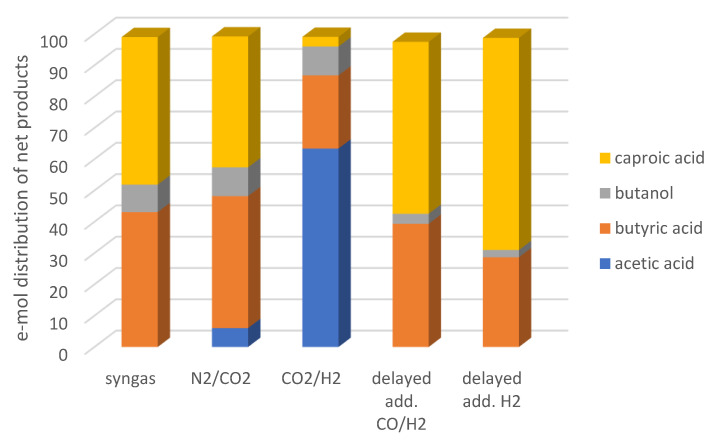
Distribution of e-mol of net major products at the end of the 1st and 3rd enrichment series.

**Figure 6 molecules-28-02562-f006:**
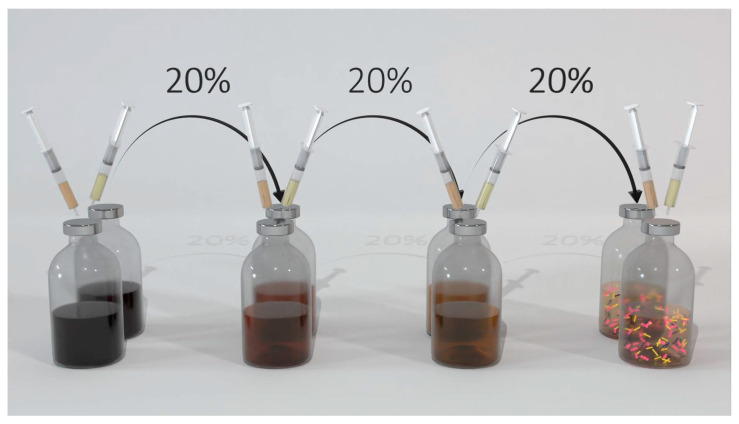
Scheme of the microbial enrichment procedure. The enrichments were performed in triplicate.

**Table 1 molecules-28-02562-t001:** Concentration of major compounds measured, and e-recovery calculated at the end of the last transfer of microbial enrichments on syngas and CO_2_/H_2_ supplemented with acetic acid and ethanol.

	1st Experimental Series on:		
Major Compounds	Acetic Acid/ Ethanol/Syngas	Acetic Acid/ Ethanol/CO_2_/N_2_	Acetic Acid/ Ethanol/CO_2_/H_2_
Acetic acid, g/L	2.76 ± 0.16	3.18 ± 0.38	5.33 ± 0.33
Ethanol, g/L	1.29 ± 0.24	1.42 ± 0.45	4.10 ± 0.28
Butyric acid, g/L	1.96 ± 0.04	2.05 ± 0.10	0.43 ± 0.16
Butanol, g/L	0.32 ± 0.03	0.24 ± 0.0	0.13 ± 0.06
Caproic acid, g/L	1.75 ± 0.17	1.65 ± 0.44	0.05 ± 0.03
e-recovery, %	87	93	99

**Table 2 molecules-28-02562-t002:** Concentration and net production of major compounds and targeted end-products measured at the end of the first phase (transfer 0) of microbial enrichments on syngas and CO_2_/H_2_ supplemented with butyric acid.

	2nd Enrichment Series on:			
Concentration of Major Compounds	Butyric Acid/ Syngas/ pH 6.5	Butyric Acid/Syngas/ pH 5.5	Butyric Acid/CO_2_/H_2_/ pH 6.5	Butyric Acid/CO_2_/ H_2_/pH 5.5
Acetic acid, g/L	0.43 ± 0.04	0.31 ± 0.1	0.17 ± 0.01	0.41 ± 0.02
Ethanol, g/L	-	-	-	-
Butyric acid, g/L	1.14 ± 0.02	1.21 ± 0.02	1.09 ± 0.01	1.08 ± 0.01
Butanol, g/L	0.006 ±0.009	0.01 ± 0.01	0.003 ± 0.005	0.01 ± 0.00
Caproic acid, g/L	-	-	-	-
e-recovery, %	78	78	75	75
**Net production**	**Butyric acid/syngas/pH 6.5**	**Butyric acid/syngas/pH 5.5**	**Butyric acid/CO_2_/H_2_/pH 6.5**	**Butyric acid/CO_2_/H_2_/pH 5.5**
Acetic acid, g/L	0.009 ± 0.002	0.078 ± 0.02	0.313 ± 0.01	0.239 ± 0.022
Butyric acid, g/L	0.041 ± 0.008	0.070 ± 0.03	−0.134 ± 0.001	−0.102 ± 0.025
Butanol, g/L	0.006 ± 0.01	0.003 ± 0.006	0.003 ± 0.006	0.013 ± 0.00

**Table 3 molecules-28-02562-t003:** Concentration of major compounds measured, and e-recovery calculated at the end of the last transfer of microbial enrichments on CO/H_2_ and H_2_ supplemented with acetic acid and ethanol.

	3rd Enrichment Series on:	
Major Compounds	Acetic Acid/ Ethanol/CO/H_2_	Acetic Acid/Ethanol/H_2_
Acetic acid, g/L	1.42 ± 0.04	1.33 ± 0.02
Ethanol, g/L	2.77 ± 0.15	2.12 ± 0.11
Butyric acid, g/L	1.04 ± 0.03	1.30 ± 0.05
Butanol, g/L	0.07 ± 0.00	0.05 ± 0.00
Caproic acid, g/L	1.51 ± 0.15	2.37 ± 0.11
e-recovery, %	87	96

**Table 4 molecules-28-02562-t004:** Growth medium composition.

	Volume Added for 1 L of Medium, mL	Components	Concentration in the Stock Solution, g/L
Macronutrients	20	NH_4_Cl	100
		NaCl	10
		MgCl_2_·6H_2_O	10
		CaCl_2_·2H_2_O	5
Yeast extract	20	Yeast extract	25
Sodium sulfate	10	Na_2_SO_4_	100
Vitamins	10	Biotin	0.002
		Folic acid	0.002
		Pyridoxine·HCl	0.01
		Riboflavin·HCl	0.005
		Thiamine·HCl	0.005
		*p*-aminobenzoic acid	0.0001
		cyanocobalamin	0.005
		nicotinic acid	0.005
		lipoic acid	0.005
Micronutrients	10	Nitrilotriacetic acid	2
		MnSO_4_·H_2_O	1
		Fe(SO_4_)_2_(NH_4_)_2_·6H_2_O	0.8
		CoCl_2_·6H_2_O	0.2
		ZnSO_4_·7H_2_O	0.2
		CuCl_2_·2H_2_O	0.02
		NiSO_4_·6H_2_O	0.02
		Na_2_MoO_4_·2H_2_O	0.02
		Na_2_SeO_3_	0.018
		Na_2_WO_4_·2H_2_O	0.022
		H_3_BO_3_	0.01
		AlCl_3_	0.01

**Table 5 molecules-28-02562-t005:** Overview of the mesophilic (37 °C) experimental series targeting C4 and C6 acids and butanol formation.

	Intermediates Added	Initial pH	Initial Gas Composition and Amounts of CO and H_2_ Added
**1st experimental series**	Acetic acid (39 mM) Ethanol (93 mM)	6.5	70% H_2_ (9.9 mmole), 10% CO (1.4 mmole), 20% CO_2_
	Acetic acid (39 mM) Ethanol (93 mM)	6.5	80% N_2_, 20% CO_2_
	Acetic acid (39 mM) Ethanol (93 mM)	6.5	72% H_2_ (10.2 mmole), 28% CO_2_
**2nd experimental series**	Butyric acid (15 mM)	6.5	72% H_2_ (10.2 mmole), 11% CO (1.9 mmole), 17% CO_2_
	Butyric acid (15 mM)	5.5	72% H_2_ (10.2 mmole), 11% CO (1.9 mmole), 17% CO_2_
	Butyric acid (15 mM)	6.5	75% H_2_ (10.6 mmole), 25% CO_2_
	Butyric acid (15 mM)	5.5	75% H_2_ (10.6 mmole), 25% CO_2_
**3rd experimental series**	Acetic acid (39 mM) Ethanol (93 mM)	6.5	67% N_2_, 22% H_2_ (3.9 mmole), 11% CO (1.9 mmole)
	Acetic acid (39 mM) Ethanol (93 mM)	6.5	78% N_2_, 22% H_2_ (3.9 mmole)

## Data Availability

Data reported in this study are available upon request.

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
