# Peer review of "Microbial Enrichment Techniques on Syngas and CO2 Targeting Production of Higher Acids and Alcohols"

_molecules, 2023, doi:10.3390/molecules28062562_

Round 1

Reviewer 1 Report

This paper describes an extended and carefully planned experimental work with interesting results and conclusions. I really enjoyed reviewing this and reading about practical methods for interpreting results, such as the SQI. In what follows, there are some comments which I hope will help to improve the good quality of this paper.

Please read carefully the text and improve the linguistic aspect. For example use commas, prefer short sentences, use the article "the" more often etc. Some examples are listed below:

Line 39: put “,” before “with reducing power”

Line 42: put “During” at the beginning of the sentence and a “,” after “decades”.

Line 44: put a “,” before “like”.

Line 55: put a “,” before “while”.

Line 70: Consider using “option” instead of “feature”.  Put a “,” after “contrary”

Line 128: put an “a” before the “mixture”

Line 129: Consider using “draw” instead of “allow”.

Line 303-305. Please rephrase, this sentence is rather complicated. Moreover, consider using “able to do so” instead of “possessing this possibility” – it does not sound good to me.

These are some examples, but there are many cases which lack commas or the article “the”.

Other comments

Line 36: explain in what way, biomass is “challenging”?

Line 46: explain the abbreviation MMC (and all other abbreviations such as DO) when referred to in the text for the first time.

Since the Methodology section is at the end because of the journal’s choice, I think there must be a bridge between the introduction and the results section. The reader misses the information of what “transfer” is as well as 1st, 2nd series etc. I would suggest a brief description like “the experimental protocol was based on microbial enrichment tests performed at different conditions, in terms of acid/alcohol and gas addition. Each enrichment experiment was performed through successive batch tests conducted by transferring a part of the mixed liquor from one batch reactor at the end of its cycle to another”. It would also be advisable to forward the readers’ attention to the sections of the methods to facilitate their understanding. I am not sure if this is the right way to do it, but it is really difficult to go from the introduction to the results which present many details based on a methodology that has not been described yet… Anyway, this is one suggestion. In the same sense, a short sentence at the beginning of 2.1, 2.2 and 2.3 describing the characteristic of each series of experiments with respect to the other two, would also help understanding.

Do you think these results are influenced by the operational mode of the reactors (that is, batch)? If the enrichments would be used in continuously run reactors, would the same results be observed? I would suggest the authors introduce this aspect (operation mode) and discuss it in their paper.

Have you an indication of the rate of these products generation (in mol per volume of reactor per day) taking the performance of the last transfer? Are these rates considered low or high in comparison with other studies? I would suggest the authors introduce this aspect (rate of production) and discuss it in their paper

Consider rephrasing the legend of figure 6 as “Scheme of the microbial enrichment procedure. The enrichment procedure was performed in triplicate.

Consider rephrasing the title of section 4.3.1. and 4.3.2 as “Microbial enrichment methodology” and “Experimental strategy”, respectively, or something similar, because the reader thinks that he/she is going to read about the same thing (microbial enrichment). Actually, I was confused by the term “series of microbial enrichment experiments”, because in my mind, by enrichment I was thinking of a series of experiments performed using a 20% volume transfer. That is, each enrichment is already a series of experiments. But when it is written “the 1st series of microbial enrichment” it meant that the microbial enrichment (which is a series of transfers) was repeated under different conditions. Therefore, in order to make the description and the discussion of the results clearer, I would call the transfers “microbial enrichment” as you do, and the different experiments “series of microbial response/ microbial adaptation/tests/experiments”. Please take this as a suggestion to improve the clarity of your paper.

Line 365: “the content of the flasks was mixed”: Do you mean that the content of all five flasks was mixed? Please be clear about this.

Line 387: “20% inoculum size”: Do you mean that 20% of the 100 ml active volume was the inoculum? Please clarify what you mean.

Line 427: “The ratio of the gases”: Do you mean the composition of the gas mixtures? Please clarify.

Author Response

Response to reviewers' comments.

We thank REVIEWER 1 for the comments and efforts to review our manuscript. We hereby provide our response - in green.

REVIEWER 1 – comments

This paper describes an extended and carefully planned experimental work with interesting results and conclusions. I really enjoyed reviewing this and reading about practical methods for interpreting results, such as the SQI. In what follows, there are some comments which I hope will help to improve the good quality of this paper.

Please read carefully the text and improve the linguistic aspect. For example use commas, prefer short sentences, use the article "the" more often etc. Some examples are listed below:

Line 39: put “,” before “with reducing power”

Line 42: put “During” at the beginning of the sentence and a “,” after “decades”.

Line 44: put a “,” before “like”.

Line 55: put a “,” before “while”.

Line 70: Consider using “option” instead of “feature”.  Put a “,” after “contrary”

Line 128: put an “a” before the “mixture”

Line 129: Consider using “draw” instead of “allow”.

Line 303-305. Please rephrase, this sentence is rather complicated. Moreover, consider using “able to do so” instead of “possessing this possibility” – it does not sound good to me.

These are some examples, but there are many cases which lack commas or the article “the”. All previous points have been corrected in the revised version. We have also carefully checked and revised the manuscript as per the reviewer’s suggestion.

Other comments

Line 36: explain in what way, biomass is “challenging”? explanation is now provided in the revised manuscript (line 36).

Line 46: explain the abbreviation MMC (and all other abbreviations such as DO) when referred to in the text for the first time. Done

Since the Methodology section is at the end because of the journal’s choice, I think there must be a bridge between the introduction and the results section. The reader misses the information of what “transfer” is as well as 1st, 2nd series etc. I would suggest a brief description like “the experimental protocol was based on microbial enrichment tests performed at different conditions, in terms of acid/alcohol and gas addition. Each enrichment experiment was performed through successive batch tests conducted by transferring a part of the mixed liquor from one batch reactor at the end of its cycle to another”. It would also be advisable to forward the readers’ attention to the sections of the methods to facilitate their understanding. I am not sure if this is the right way to do it, but it is really difficult to go from the introduction to the results which present many details based on a methodology that has not been described yet… Anyway, this is one suggestion. In the same sense, a short sentence at the beginning of 2.1, 2.2 and 2.3 describing the characteristic of each series of experiments with respect to the other two, would also help understanding. We fully agree with the reviewer and have added critical explanations in the revised manuscript to facilitate understanding. See lines 135-139, 142-145, 172-175, 203-209.

Do you think these results are influenced by the operational mode of the reactors (that is, batch)? If the enrichments would be used in continuously run reactors, would the same results be observed? I would suggest the authors introduce this aspect (operation mode) and discuss it in their paper. The reviewer very correctly pointed out the importance of the operation mode. We fully agree and we have now addressed this issue in the discussion section, lines 310-314.

Have you an indication of the rate of these products generation (in mol per volume of reactor per day) taking the performance of the last transfer? Are these rates considered low or high in comparison with other studies? I would suggest the authors introduce this aspect (rate of production) and discuss it in their paper Unfortunately, we cannot draw any conclusions on the production rates; we focused on products distribution and yields and therefore the sampling protocol does not allow us to extract rates. 

Consider rephrasing the legend of figure 6 as “Scheme of the microbial enrichment procedure. The enrichment procedure was performed in triplicate.” Done.

Consider rephrasing the title of section 4.3.1. and 4.3.2 as “Microbial enrichment methodology” and “Experimental strategy”, respectively, or something similar, because the reader thinks that he/she is going to read about the same thing (microbial enrichment). Actually, I was confused by the term “series of microbial enrichment experiments”, because in my mind, by enrichment I was thinking of a series of experiments performed using a 20% volume transfer. That is, each enrichment is already a series of experiments. But when it is written “the 1st series of microbial enrichment” it meant that the microbial enrichment (which is a series of transfers) was repeated under different conditions. Therefore, in order to make the description and the discussion of the results clearer, I would call the transfers “microbial enrichment” as you do, and the different experiments “series of microbial response/ microbial adaptation/tests/experiments”. Please take this as a suggestion to improve the clarity of your paper. The titles of the sections have been changed considering the suggestion of the reviewer. We also thank the reviewer for the recommendation to clarify experimental series versus enrichment series. Following his/her suggestion we have revised the terminology we use to clearly differentiate the different “series” in our work.

Line 365: “the content of the flasks was mixed”: Do you mean that the content of all five flasks was mixed? Please be clear about this. We have clarified this in the text – line 389.

Line 387: “20% inoculum size”: Do you mean that 20% of the 100 ml active volume was the inoculum? Please clarify what you mean. The corresponding text is now revised to facilitate understanding – lines 379-380.

Line 427: “The ratio of the gases”: Do you mean the composition of the gas mixtures? Please clarify. We have replaced “ratio of the gases” with “composition of the gas mixtures” – line 450 in the revised manuscript.

Reviewer 2 Report

The manuscript molecules-2236637 deals with the production of higher acids/alcohols by performing microbial enrichment techniques on syngas and CO2. In particular,  the enrichment technique yielded a maximum increase of 43 and 68% for butyric and caproic acid, respectively, when acetic acid and ethanol were supplementated together with the gas substrates. Butanol formation was also enhanced up to 9%. 

The methodology is appropriately described and the results well presented. Therefore the manuscript can be accepted for publication after a minor spell check.

Author Response

We thank REVIEWER 2 for reviewing our manuscript and for the kind words. We have spell-checked the manuscript as suggested.